# Herpes zoster epidemiology in Latin America: A systematic review and meta-analysis

**Ariel Esteban Bardach**[ID][1]*, **Carolina Palermo**[1], **Tomás Alconada**[1], **Macarena Sandoval**[1], **Darío Javier Balan**[1], **Javier Nieto Guevara**[2], **Jorge Gómez**[ID][2], **Agustin Ciapponi**[ID][1]

1 Institute for Clinical Effectiveness and Health Policy, Buenos Aires, Argentina, 2 GSK, Buenos Aires, Argentina

* abardach@iecs.org.ar

**Data Availability Statement:** All relevant data are within the manuscript and its Supporting Information files.

## Abstract

The epidemiology and burden of Herpes Zoster (HZ) are largely unknown, and there are no recent reviews summarizing the available evidence from the Latin America and Caribbean (LAC) region. We conducted a systematic review and meta-analysis to characterize the epidemiology and burden of HZ in LAC. Bibliographic databases and grey literature sources were consulted to find studies published (January 2000 –February 2020) with epidemiological endpoints: cumulative incidence and incidence density (HZ cases per 100,000 person-years), prevalence, case-fatality rates, HZ mortality, hospitalization rates, and rates of each HZ complication. Twenty-six studies were included with most studies coming from Brazil. No studies reported the incidence of HZ in the general population. In population at higher risk, the cumulative incidence ranged from 318–3,423 cases of HZ per 100,000 persons per year of follow-up. The incidence density was 6.4–36.5 cases per 1,000 person-years. Age was identified as a major risk factor towards HZ incidence which increase significantly in people >50 years of age. Hospitalization rates ranged from 3%–35.7%. The in-hospital HZ mortality rate ranged from 0%–36%. Overall, HZ mortality rates were found to be higher in females across all age groups and countries. The incidence of HZ complications (such as post-herpetic neuralgia, ophthalmic herpes zoster, and Ramsay Hunt syndrome) was higher in the immunosuppressed compared to the immunocompetent population. Acyclovir was the most frequently used therapy. Epidemiological data from Ministry of Health databases (Argentina, Brazil, Colombia, Chile y Mexico) and Institute for Health Metrics and Evaluation's Global Burden of Disease project reported stable rates of hospitalizations and deaths over the last 10 years. High-risk groups for HZ impose a considerable burden in LAC. They could benefit from directed healthcare initiatives, including adult immunization, to prevent HZ occurrence and its complications.

## Introduction

Primary exposure to varicella zoster virus (VZV) manifests as chickenpox, usually in children, afterwards the virus has the ability to remain dormant in the infected individual [1]. A decrease in cell-mediated immunity can reactivate VZV which causes herpes zoster (HZ), commonly known as shingles, in adults [2]. Factors associated with the reactivation of dormant VZV

**Funding:** This study was supported by GlaxoSmithKline Biologicals SA in the form of grants awarded to AB, AC, TA, CP, MS, and DB, salaries for JG and JN, and all costs associated with the development and publication of this article. The specific roles of these authors are articulated in the 'author contributions' section. GlaxoSmithKline Biologicals SA had a role in study design, data collection and analysis, decision to publish, and preparation of the manuscript.

**Competing interests:** The authors have read the journal's policy and have the following competing interests: AB, AC, TA, CP, MS, and DB received grant support from GlaxoSmithKline Biologicals SA, which funded the development and publication of this article. JG and JN are employees of GlaxoSmithKline Biologicals SA. This does not alter our adherence to PLOS ONE policies on sharing data and materials. There are no patents, products in development or marketed products associated with this research to declare.

include immune-suppression, aging, intra-uterine exposure to VZV and, exposure to VZV before 18 months of age [1]. Other risk factors include gender, ethnicity, family history, and co-morbidities like asthma, diabetes, systemic lupus and other chronic pulmonary diseases [3,4]. The lifetime risk of contracting HZ is estimated to be 15%-30%, and the risk is higher in older adults, immunocompromised individuals, and those with underlying comorbid conditions [5,6], for whom the disease is more severe and the likelihood of complications is also higher.

HZ manifests itself as a unilateral painful skin rash with blisters [7,8], and usually extends for about 7–10 days, with the rash typically healing within 2–4 weeks; in severe cases, HZ causes pain which may last for several months to a few years [7,9]. Post-herpetic neuralgia (PHN), defined as pain that persists for at least three months since the onset of the rash [10], is the most common complication of HZ and occurs in about one in five patients [11]. The duration of PHN could last for weeks, months, or even years. The quality of pain varies from mild to severe, constant, or intermittent and, may be initiated by trivial stimuli [4]. The recurrent pain disrupts sleep, daily activities and the ability to work, thus leading to reduced quality of life and a depressed state of mind [12]. Other complications associated with HZ include stroke or other cardiovascular events, neurological sequelae, palsy and gastrointestinal ailments [13]. Severe cases of the above complications often require hospitalization [13].

Prompt antiviral therapy is the recommended treatment for HZ patients, preferable within 72 hours of onset of the rash. Immunocompromised and other high-risk individuals may be given intravenous antiviral medication [2]. The antiviral drug of choice for uncomplicated HZ is an antiviral therapy with oral acyclovir, valacyclovir, or famciclovir [14]. At present, the efficacy of the antiviral treatment to prevent PHN is unclear [15,16]. Management of PHN pain is highly complex, depends on the patient's pain characteristics, and is generally inadequate [16,17]. Treatment ranges from topical agents (lidocaine or capsaicin), anticonvulsants (gabapentin, pregabalin) to antidepressants (tricyclic antidepressants) [12,18].

To prevent HZ and PHN, the Zoster Vaccine Live (ZVL), a single-dose subcutaneous live attenuated vaccine, was first introduced and approved by the Food and Drugs Administration (FDA) for use in adults over 60 years and in adults over 50 years in 2006 and 2011, respectively [19]. Thereafter, it was approved in countries such as Australia, Canada and different countries in Europe and Latin America such as Colombia, Chile, Mexico and Venezuela. In Argentina it has been available since 2014 and was authorized by the Administración Nacional de Medicamentos, Alimentos y Tecnología Médica (ANMAT) to be used in people over 50 years of age. Since 2017, a two-dose Recombinant Zoster Vaccine (RZV), has also become available in the United States for use in immunocompetent adults >50 years of age. Patients with prior ZVL are advised to receive the two doses of RZV due to its higher efficacy and length of protection and to prevent both HZ and PHN (particularly in people >70 years of age). While the best time to repeat vaccination is not fully clear, the United States Centers for Disease Control and Prevention (CDC) Advisory Committee on Immunization Practices (ACIP) recommends administering the first dose of RZV at least eight weeks after receiving ZVL [20].

An annual rate of new HZ cases ranging from 3–5 cases per 1,000 inhabitant-years has been reported in North America, Europe, and Asia-Pacific, with few data available from the regions of Africa, Asia and the Latin America and Caribbean (LAC) region [21]. Moreover, it has been observed that the incidence of HZ increases with increasing age. Considering age is a risk factor for HZ and its associated complications, the increasing life expectancy in the general population may considerably increase HZ annual cases and disease burden [22]. The epidemiology and burden of HZ are largely unknown for the LAC region, and there are no recent reviews summarizing the available evidence from the LAC region. A previous review conducted for the LAC region found evidence on HZ disease and outcomes from only three studies [5]. This

is probably due to lack of mandatory reporting and surveillance. However, such data are crucial to ensure that existing disease control and prevention policies are still pertinent and if not, informed decisions about such policies can be implemented in all countries of the LAC region.

The objective of this review was to describe the epidemiology of HZ on individuals ≥15 years of age, particularly age-based incidence, length of acute disease and frequency of complications in the LAC region. To achieve this objective, we conducted a systematic literature review to collect previously published information on the epidemiology and burden of HZ in the LAC region in the last 20 years, considering incidence, prevalence, morbidity, and mortality.

## Methods

We performed a systematic review of the literature following the Cochrane Systematic Reviews Manual [23] and Preferred Reporting Items for Systematic Literature Reviews and Meta-Analyses (PRISMA) [24,25]. In addition, we followed the Meta-analyses of Observational Epidemiology (MOOSE) guidelines specifically for reviews of observational trials [26]. The protocol is registered with prospective international systematic review registry PROSPERO (CRD42020186586) [27].

### Search sources and strategy

We searched the following online databases: PubMed, Latin American and Caribbean Health Sciences Literature (LILACS), Excerpta Medica Database (EMBase), Cumulative Index of Nursing and Allied Health Literature (CINAHL), Cochrane Library, Centre for Reviews and Dissemination (CRD) York, and EconLIT for eligible studies. The search was conducted using both the indexed word and the keywords in the title and abstract (**S1 Text in S1 File**). We combined the search terms using Boolean operators for different databases. Searches were limited to capture articles published between 01 January 2000 and 20 February 2020.

Manual searches were performed across lists of references from any papers included to obtain further information. Databases containing national and international congresses proceedings and doctoral theses were consulted. The websites of major local medical associations, experts, and associations related to the field were searched, and the authors of relevant papers were inquired about any missing or clarifying information.

In addition, grey literature such as websites of local Departments of Health, the Pan American Health Organization, the Virtual Health Library, and hospital reports was examined. The Department of Health websites from the LAC countries were searched to retrieve data on hospitalization and mortality associated with HZ. Ministerial websites referring to epidemiological data and the burden of HZ were assessed according to the inclusion criteria specified. The Global Burden of Disease (GBD) database was assessed for information on mortality rates in varicella and HZ patients and on the incidence of HZ mortality rates from the period of 2010–2017 in most LAC countries [28].

### Article selection and data extraction

Relevant publications were identified independently by peer review utilizing the inclusion and exclusion criteria provided in **Table 1**. Discrepancies were solved with the agreement of the entire team. All screening phases of the study used COVIDENCE® [29,30], a web-based platform designed to process systematic reviews.

From the list of eligible articles, the research team extracted data based on three pre-defined parameters: publication and study characteristics (type of publication, year published, authors, geographic location, study design including domains for the risk of bias method), study

**Table 1. Inclusion and exclusion criteria.**

| | Inclusion criteria | Exclusion criteria |
|---|---|---|
| Population | ▪ Participants ≥15 years of age:<br>○ Average-risk: Susceptible to HZ or with a likely or confirmed case of HZ<br>○ High-risk individuals between 50–60 years of age, trauma, those with a transplantation, diagnosed with HIV/AIDS, cancer, under treatment with corticosteroids/immunosuppressants/chemotherapy. | ▪ Populations outside the scope of the inclusion criteria |
| Intervention | ▪ Not restricted by intervention | ▪ Not applicable |
| Comparator | ▪ Not restricted by comparator | ▪ Not applicable |
| Outcome | ▪ Cumulative incidence and incidence density (HZ cases per 100,000 person-years)<br>▪ Prevalence<br>▪ Case-fatality rates and cause-specific mortality<br>▪ Hospitalization (rates, lenght of stay, discharges)<br>▪ Rates of HZ complications | ▪ All other outcomes than those specified as eligible |
| Study design | ▪ Randomized and non-randomized epidemiological studies*<br>○ Randomized studies (meeting EPOC criteria [23])<br>• Randomized controlled trials (control arms)<br>• Quasi-randomized trials<br>• Controlled before-after and uncontrolled before-after studies<br>• Interrupted time series including controlled interrupted time series<br>○ Observational studies<br>• Cohort studies<br>• Case-control studies<br>• Cross-sectional studies<br>• Ecological study<br>• Case series (involving at least 50 HZ cases and 10 HZ complication)<br>▪ Epidemiological surveillance reports | ▪ Systematic reviews**<br>▪ Meta-analyses**<br>▪ Narrative reviews (without methods)<br>▪ Interventional studies<br>○ Randomized studies<br>○ Non-randomized studies<br>▪ Cost-effectiveness or health economics studies<br>▪ Surveys<br>▪ Non-human data (e.g. animal models, in-vitro, in-silico) or predictions via modeling methods<br>▪ Case reports<br>▪ Letter to editor<br>▪ Newspaper<br>▪ Editorial<br>▪ Comment<br>▪ Opinions<br>▪ Molecular studies<br>▪ Pilot studies<br>▪ Protocols/pre-clinical studies<br>▪ Studies with insufficient methodological details |
| **Limits** | | |
| Publication date | 01 January 2000 to 20 February 2020 | All publications outside the eligible time period |
| Geographic scope | Latin America and Caribbean region | All other countries |
| Language | English, Spanish, Portuguese | All other languages |

EPOC: Effective Practice and Organization of Care; HZ: Herpes zoster.

*References cited by screened articles were manually reviewed for relevance (i.e. snowballing).

**References of included articles in these systematic reviews/meta-analyses were manually screened for additional relevant original articles (as deemed necessary by the reviewer).

population characteristics (age, sex, sample size, latent immune-compromising conditions, risk evaluation for HZ, inclusion and exclusion criteria), and outcomes (rate of incidence, HZ mortality, fatality rate, rate of hospitalization, length and recurrence of an acute episode and disease complications). The authors of the publication were contacted when necessary to get any missing or clarifying information. For data or subsets of data reported more than once, the one with the largest sample size was selected.

## Risk of bias assessment

The risk of bias assessment was performed independently by at least two reviewers, and discrepancies were resolved in consensus with the whole team. Different approaches for risk of

bias assessment were taken depending on the study design. For observational studies, the risk of bias assessment was according to the United States National Heart, Lung and Blood Institute guidelines checklists [31]. The studies were rated as "Bad" for a high-risk of bias, "Poor" for an uncertain risk of bias, and "Good" for a low risk of bias. A total of 14 and 9 items were analyzed to assess the risk of bias in cohort and cross-sectional studies, and case series, respectively.

## Statistical analyses and reporting

In this paper, we provide a descriptive overview of epidemiological outcomes from published literature, government databases and the Institute for Health Metrics and Evaluation (IHME) GBD project database, as of 2019. We also present the risk of bias results using a chart and, when relevant, a summary of findings was also included.

In order to analyze data, we performed a meta-analysis of proportions when it was considered methodologically appropriate. An arcsine transformation was applied to stabilize variance in proportions (Freeman-Tukey variance of the arcsine square root transformation of proportions), where y = arcsine $[\sqrt{(r/(n + 1))}]$ + arcsine $[\sqrt{(r/(n + 1)/(n + 1))}]$, with a variance of $1/(n + 1)$, where n is the population size [32]. Pooled proportion was estimated as transformation of the weighted means of transformed proportions using inverse arcsine variance weights for the fixed and randomized effects model. In order to evaluate consistency of this method, generalized linear mixed models were used [33]. DerSimonian-Laird weights were applied to the randomized effects model [34]. In case of heterogeneity across studies, the statistics were estimated as a measure of the general variance proportion attributable to heterogeneity across studies, and its causes were explored using subgroups and sensitivity analyses [35]. Statsdirect [36] and STATA 15.0 [37] were used for all analyses. When the follow-up period across studies changed considerably, rates of incidence in person-years were estimated by dividing the number of new cases occurred during the follow-up period (numerator) by the total person-time units (person-years) from the risk group (denominator). The rate of incidence in the person-time or rate of density of incidence is considered an appropriate incidence measure when follow-up times are different [38]. Incidence was expressed as the number of cases per 100,000 person-years.

When allowed by the number of studies, we planned to perform subgroup analyses based on the study design, five-year data, country, sex and age group (15–64 years and ≥65 years), risk level of the population (high-risk [immunosuppressed] and normal risk), and country rating (based on the World Bank per capita income, [low, mid-low, mid-high, high]) [39]. We also planned to perform sensitivity analyses to evaluate the impact of the risk of bias on the results of primary analyses, restricting analysis to studies with a low risk of bias for the main domains.

## Results

The literature search returned 1,309 studies; of these 102 were selected for full-text review. After full-text screening, 26 studies were included in this review (**Fig 1**). The selected studies from Latin America included data from the following countries: Brazil (n = 9), Argentina (n = 5), Chile (n = 2), Colombia (n = 2), Costa Rica (n = 1), Mexico (n = 2), Nicaragua (n = 1) and Peru (n = 1). In addition, three studies which had patients represented from different Latin American countries (Argentina, Brazil, Chile, Colombia, Costa Rica, Dominican Republic, Mexico, Peru, and Venezuela) were included (**Table 2**). None of the included studies mention the vaccination status of participants.

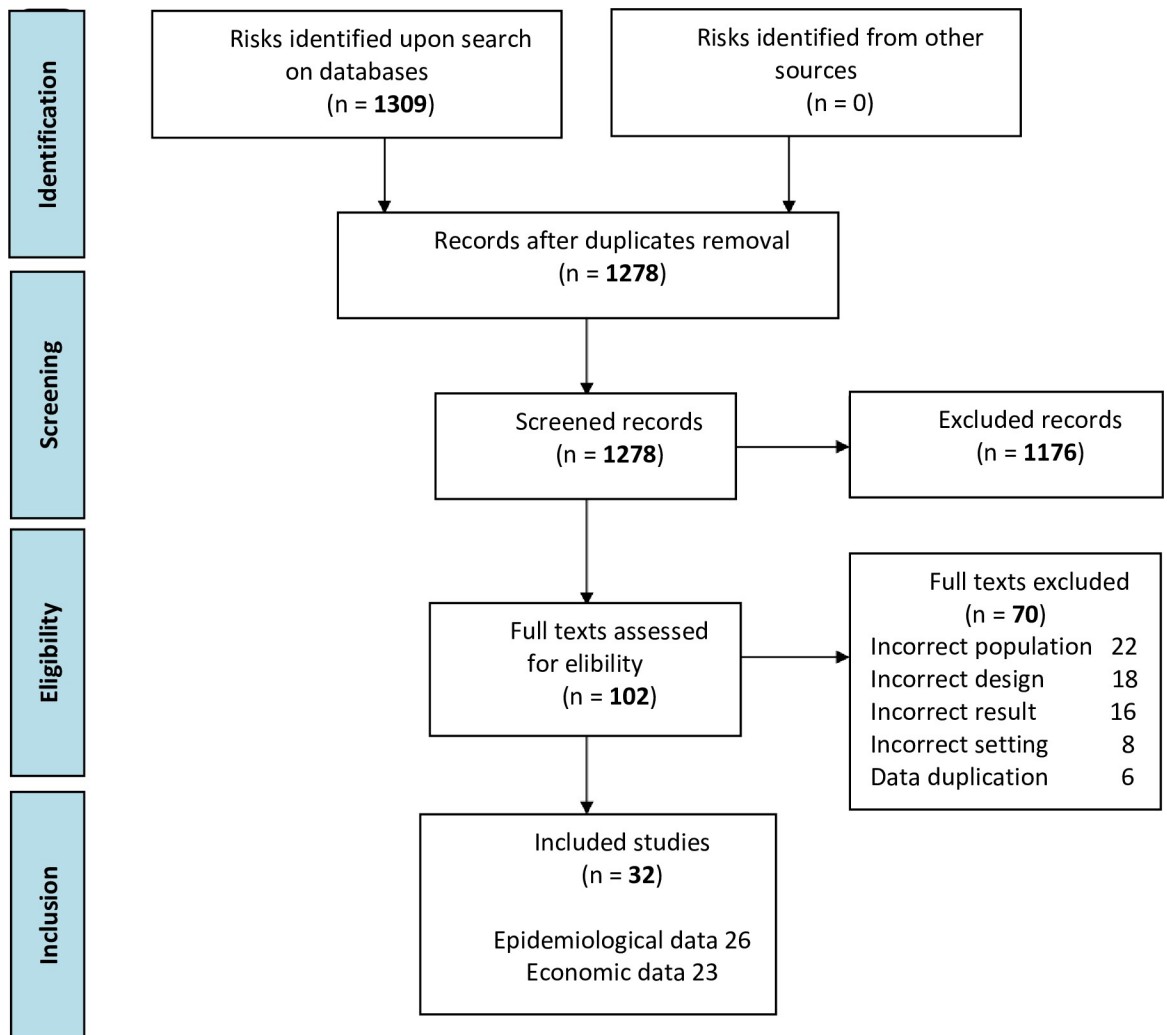

**Fig 1. PRISMA flow diagram.**

Within the 26 studies, 15 studies were case series, 9 were cohort studies and two were epidemiological surveillance studies. The reported lenght of follow-up ranged from 4 to 359 months. The majority of studies were done in outpatients (n = 11), followed by hospitalized patients (n = 5) or a mix of both (n = 2). The remaining eight studies did not have information about the source of the study population. The majority of the patients covered in the studies belonged to high-risk groups (n = 18, 56%), and most included all age groups (n = 16, 50%). The participants' age ranged from 1 to 100 years old, with a mean age ranging from 19 to 77 years old and a median age ranging from 37 to 70 years old. The female to male ratio of HZ cases was 1.1/1 with 43% of the cases documented in men. The number of HZ cases ranged from 11 to 7,042 (**Table 2**).

## Risk of bias assessment

The risk of bias assessment revealed that 73% (n = 8/11) of cohort studies and 93% (n = 14/15) of case series studies had a low risk of bias. The most challenging issues came from the assessment of cohort studies and included those relative to the rationale for sample size (100% of

**Table 2. Study and population characteristics.**

| Reference | Study design | Sample type | Study period (dd/mm/yy) | Risk of HZ | Age (range in years) | Age (mean) ±SD | Median (IQR) | Males (n) | Source population (n) | HZ cases (n) |
|---|---|---|---|---|---|---|---|---|---|---|
| **Argentina (n = 5)** | | | | | | | | | | |
| Bollea-Garlatti 2017 [40] | Series of cases | Hospitalized | 1/2/10-31/10/15 | High | All ages | NR | 70(52–82) | 21 | 41 | 41 |
| Corti 2015 [41] | Series of cases | Hospitalized | 1/1/02-30/11/14 | High | 15–65 | 35.27±9.22 | 37 | 11 | 11 | 11 |
| Rozenek 2018 [42] | Series of cases | Outpatient | 1/6/13-31/5/17 | High | >65 | 76.77 | NR | 403* | 1,267 | 1,267 |
| Vujacich 2008 [43] | Series of cases | Outpatient | 1/1/00-31/12/05 | Normal | All ages | NR | 57(33–69) | 113 | 302 | 302 |
| Vujacich 2016 [44] | Cohort | Outpatient | 1/7/07-30/9/12 | High | >65 | 70±10.7 | NR | 32 | 96 | 96 |
| **Brazil (n = 9)** | | | | | | | | | | |
| Antoniolli 2019 [45] | Series of cases | Hospitalized | 1/3/00-31/1/17 | High | All ages | 48.8±22.2 | NR | 353 | 801 | 801 |
| Álvarez 2007 [46] | Series of cases | Outpatient | 1/8/04-30/11/04 | Normal | All ages | 72.11 | NR | 4 | 18 | 18 |
| Andrade 2019 [39] | Series of cases | Outpatient | 1/3/14-31/10/15 | Normal | All ages | 71 | NR | 5 | 19 | 19 |
| Borba 2010 [47] | Cohort | NR | 1/1/99-30/6/06 | High | NR | 39±13.7 | NR | 5 | 1,145 | 51 |
| Carvalho 2016 [48] | Cohort | NR | 1/1/09-31/1/16 | High | NR | 51±14.3 | NR | NR | 2,715 | 61 |
| de Martino Mota 2016 [49] | Series of cases | Hospitalized | 1/1/08-31/12/13 | Normal | All ages | NR | NR | NR | NR | NR |
| Gormezano 2015 [50] | Series of cases | NR | NR | High | NR | NR | NR | 8 | 1,830 | 70 |
| Teive 2008 [51] | Series of cases | NR | 1/1/89-31/12/06 | High | All ages | 19.2 | NR | 162 | 1,000 | 270 |
| Toniolo-Neto 2018 [52] | Cohort | Outpatient | 1/5/08-31/10/09 | High | >65 | 69.9±10.9 | NR | 52 | 146 | 146 |
| **Chile (n = 2)** | | | | | | | | | | |
| Cortés 2008 [53] | Cohort | NR | 1/4/01-31/12/04 | High | NR | NR | NR | NR | 2,050 | 110 |
| Wageman 2014 [54] | Series of cases | Outpatient | 1/1/84-31/12/13 | Normal | All ages | NR | NR | 78 | 4,360 | 180 |
| **Colombia (n = 2)** | | | | | | | | | | |
| Alarcón 2014 [55] | Cohort | Mixed | 1/2/05-30/11/11 | High | All ages | NR | 54.2 | NR | 1,268 | 76 |
| Rampakakis 2019 [56] | Cohort | Outpatient | 1/11/15*-31/5/17 | Normal | All ages | 65.6±9.6 | NR | 58 | 154 | 154 |
| **Costa Rica (n = 1)** | | | | | | | | | | |
| Rampakakis 2017 [57] | Cohort | Mixed | 11/1/08-30/6/10 | High | >65 | 69.5±10.8 | NR | 16 | 50 | 50 |
| **Mexico (n = 2)** | | | | | | | | | | |
| González 2013 [58] | Series of cases | Outpatient | NR | Normal | All ages | 49.8 | NR | 11 | 19 | 19 |
| Vázquez 2017 [59] | Series of cases | Hospitalized | 1/1/00-31/12/13 | High | >65 | NR | NR | 3,062 | 7,042 | 7,042 |
| **Nicaragua (n = 1)** | | | | | | | | | | |
| Mendoza Rodríguez 2007 [60] | Series of cases | Outpatient | 1/1/02-31/12/06 | Normal | All ages | NR | NR | 287 | 614 | 614 |
| **Peru (n = 1)** | | | | | | | | | | |
| Rueda 2010 [61] | Series of cases | Outpatient | 1/1/02-31/12/06 | Normal | All ages | 54 | NR | 455 | 816 | 816 |
| **Multiple countries (n = 3)** | | | | | | | | | | |
| Castañeda 2017 [62] | Cohort | NR | NR | High | All ages | 48.7 | NR | NR | 984 | 69 |

(*Continued*)

**Table 2.** (Continued)

| Reference | Study design | Sample type | Study period (dd/mm/yy) | Risk of HZ | Age (range in years) | Age (mean) ±SD | Median (IQR) | Males (n) | Source population (n) | HZ cases (n) |
|-----------|--------------|-------------|-------------------------|------------|----------------------|----------------|--------------|-----------|------------------------|--------------|
| Kawai 2015 [63] | Cohort | NR | NR | High | >65 | NR | NR | 44 | 132 | 132 |
| Zerbini 2016 [64] | Cohort | NR | NR | High | All ages | NR | NR | NR | 1,035 | 96 |

IQR: Interquartile range, NR: Not reported, SD: Standard deviation.

*Additional data provided by the author.

studies failed to do or to report this) and the statistical analyses to control potential confounders (not done or not reported in 73% of studies). In general, domains used in the assessment of case series showed no problems, but the one regarding intervention/exposure definition (item 5) was not applicable to the study objective (**S2 Table** in **S1 File**).

**Epidemiology and burden of disease.** No studies with information on the incidence of HZ for the general population were identified in this review. In populations at higher risk, the incidence density ranged from 6.4 to 36.5 cases of HZ every 1,000 patient-years while the cumulative incidence ranged from 318 to 3,423 cases of HZ per 100,000 people per year of follow-up. In the same population, the case-fatality rate (CFR) ranged from 0% to 36%. The later percentage corresponds to meningoencephalitis in the context of AIDS patients with HZ. The total number of cases of recurrent disease went from 1 to 4, with a frequency of recurrence from 1.64% [48] to 7.84% [47] in the risk population as compared to 0.16% [60] in the average-risk population, suggesting that the risk of recurrence in immunosuppressed patients is ten times higher than in the average-risk population. The mean lenght of the acute episode ranged from 10 to 29.9 days. The number of patients requiring hospitalization ranged from 4 to 7.04 with a frequency of hospitalization from 3% to 35.7% in all cases in patients from the risk group. Only two studies reported the use of laboratory diagnostic testing to confirm a HZ case (**Table 3**). No meta-analyses of incidence, hospitalization rates or CFR were possible due to high heterogeneity.

**Complications and use of antivirals.** The occurrence of PHN was reported in 15 studies, with number of patients ranging from 10 to 775. The proportion of HZ patients with PHN ranged from 11% - 22.86% in the high-risk population to 12.9%-21.82% in the average-risk population. The mean lenght of PHN, reported in one study, was 52.9 months. The proportion of patients with ophthalmic herpes zoster (OHZ) oscillated between 0% and 7.6% in high-risk patients and was 2.0% in average-risk ones. The percentages of patients with Ramsay Hunt syndrome (RHS) was 1.75% in one large series from Brazil [45]. The proportion of patients with RHS was in one study and was 1.8% in high-risk patients. Neurological complications included meningoencephalitis (5.4% in HZ inpatients, according to a large Mexican study [59]. Gonzales [68] reported 2 cases of focal motor weakness in 19 in a series of Mexican outpatients from a dermatology center. The proportion of patients with meningoencephalitis went from 2.8%–5.4% in patients with an increased risk. The frequency of secondary bacterial infection ranged from 5 to 7.5% in two large case series [67,69], while the proportion of patients with secondary bacterial infection went from 11.8%–12.9% in the population at higher risk and was 5%–7.5% in the average-risk population. The proportion of patients with disseminated HZ (DHZ) varied from 0 to 16% in the population with an increased risk and from 0.5%–7.6% in average-risk patients (**S3 Table** in **S1 File**).

Acyclovir was the most frequently prescribed treatment, administered either orally (71.7%) or intravenously (11%). The method of administration was not reported in 17.4% of patients. Other drugs prescribed were valacyclovir (6%), brivudine (0.6%), and famciclovir (0.5%). For

**Table 3. Epidemiology and burden of HZ in LAC.**

| Reference | Incidence density (number of cases per 1,000 patient-years (95% CI) | Cumulative incidence (number of cases per 100,000 population) at the end of follow-up | Deaths (number) | CFR (%) | HZ recurrence (number) | Lenght, in days, of the acute HZ episode (mean ± SD) | Hospitalizations (number) | Laboratory confirmation (number) | Method used for microbiological confirmation |
|---|---|---|---|---|---|---|---|---|---|
| Bollea-Garlatti 2017 [40] | NR | NR | 6 | 15 | NR | NR | NR | 41 | PCR or IFD |
| Corti 2015 81 [41] | NR | NR | 4 | 36 | NR | NR | 11 | 11 | PCR |
| Rozenek 2018 [42] | NR | NR | NR | NR | NR | 17.5 (3.5) | 38 | NR | NR |
| Antoniolli 2019 [45] | NR | NR | 5 | 0.62 | NR | NR | NR | NR | NR |
| Álvarez 2007 [46] | NR | NR | NR | NR | NR | 29.9 | NR | NR | NR |
| Borba 2010 [47] | 6.4 | 4.45 | NR | NR | 4 | NR | 9 | NR | NR |
| Carvalho 2016 [48] | 8.3 | 2.25 | NR | NR | 1 | NR | NR | NR | NR |
| de Martino Mota 2016 [49] | NR | NR | NR | NR | NR | NR | NR | NR | NR |
| Gormezano 2015 [50] | NR | NR | 0 | 0 | 4 | 10 (5–30) | 25 | 0 | NR |
| Toniolo-Neto 2018 [52] | NR | NR | NR | NR | NR | NR | 13 | NR | NR |
| Cortés 2008 [65] | NR | 5.36 | NR | NR | NR | NR | NR | NR | NR |
| Alarcón 2014 [66] | NR | 6.02 (4.80–7.0) | NR | NR | NR | NR | NR | NR | NR |
| Rampakakis 2017 [57] | NR | NR | NR | NR | NR | NR | 16 | NR | NR |
| Castañeda 2017[62] | 33.9 (26.8–42.9) | 7.01 | NR | NR | NR | NR | NR | NR | NR |
| Zerbini 2016 [64] | 36.5 (29.6–44.6) | 9.28 | NR | NR | NR | NR | NR | NR | NR |
| Vázquez 2017 [59] | NR | NR | NR | NR | NR | NR | 7,042 | NR | NR |
| Mendoza Rodríguez 2007 [67] | NR | NR | NR | NR | 1 | NR | NR | NR | NR |

CI: Confidence interval, CFR: Case-fatality rate, HZ: Herpes zoster, IFD: Direct immunofluorescence, LAC: Latin America and Caribbean; NR: Not reported, PCR: Polymerase chain reaction, SD: Standard deviation.

*Additional data provided by the author.

pain management, patients received non-steroid anti-inflammatory drugs (NSAIDs), anticonvulsants, tricyclic antidepressants, and topical pain-relieving drops. Seventy-one patients reported using corticosteroids, and 14 patients reported use of antibiotics. In addition, other types of treatments were reported, predominantly topical medication (referred to as drying agents, antivirals, antibiotics, steroids and/or analgesics) in 570 patients. Finally, there are no data on prophylactic treatment, prophylactic effects, or secondary effects related to the medication prescribed (**S3 Table in S1 File**).

**Occurrence of complications in higher- and average-risk patients.** A meta-analysis of proportions was conducted for HZ complications in two subgroups, immunocompromised (high-risk) and immunocompetent (average-risk) patients. The random effects model was used for this meta-analysis to address the high heterogeneity found in studies, mainly explained by different population sources. The pooled proportion of PHN in immunosuppressed patients and the immunocompetent population was 22% (95% CI = 20%-25%; $I^2$ = 0%) and 16% (95% CI = 12%-21%; $I^2$ = 94%), respectively. Three studies (Rampakakis 2017, Rampakakis 2019 [56] and Vujacich 2016 [44]) were excluded from the meta-analysis, since all eligible patients included were those with pain associated with HZ (acute neuritis or PHN). Regarding OHZ, a pooled proportion of 3.4% was found in immunosuppressed patients (95% CI = 0.7% to 8%; I2 = 77.6%). In the immunocompetent population, a meta-analysis on this complication was not feasible due to high heterogeneity. The occurrence of secondary bacterial infection was two times higher in the immunosuppressed population (13%; 95% CI: 7.6%-20%; $I^2$ = 0%) than in the immunocompetent population (6.5%; 95% CI: 4.3%-9%; $I^2$ = 51.5%). A moderate level of heterogeneity was documented due to large differences among studies in the number of patients. With regards to the occurrence of DHZ, the proportion was higher in the immunosuppressed population (4.5%; 95% CI: 0.06%-19%; $I^2$ = 94.9%). In the immunocompetent population, the pooled proportion of DHZ was 2.9% (95%CI = 0.8%-6.4%; $I^2$ = 94.5%) (**Fig 2**).

**Epidemiological data from government databases.** *Brazil.* Health information published by the Information Technology Department from the Unified Health System (DATA-SUS) was explored [70]. The DATA-SUS system consolidates information received from different hospital units. Regarding HZ hospitalizations, a clear-cut distinction could not be made between in-hospital deaths attributable to HZ or those due to varicella, because of the same

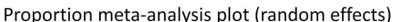

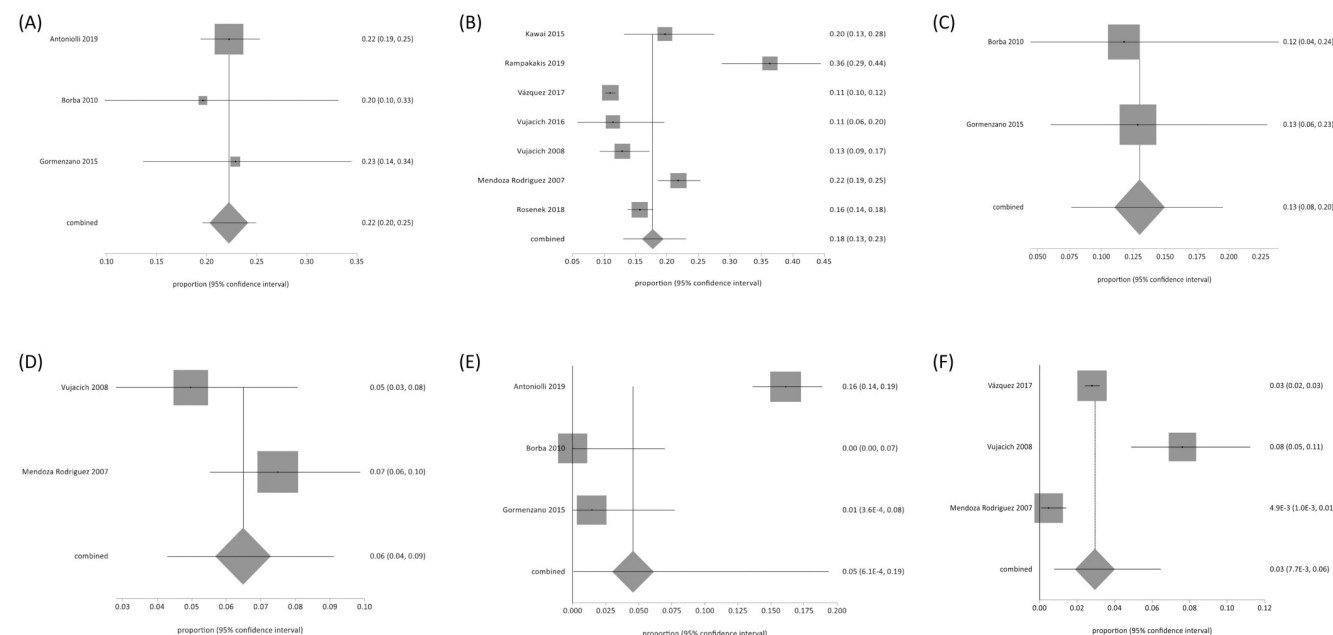

**Fig 2. HZ complications in immunosuppressed patients versus immunocompetent patients.** (A) PHN in immunosuppressed patients (B) PHN proportion in immunocompetent patients (C) Secondary bacterial infection in immunosuppressed patients (D) Secondary bacterial infection in immunocompetent patients (E) DHZ in immunosuppressed patients (F) DHZ in immunocompetent patients.

coding. Hence, we assumed that hospitalizations for patients ≥65 years of age were mostly due to HZ [48]. From 2010 to 2019, there were 16,617 hospitalizations caused by VZV in patients ≥65 years of age. Total annual hospitalizations ranged from 1,381 (in 2014) to 1,974 (in 2010), and globally involved 119,562 days in the hospital (ranging from 9,746 in 2012 to 13,497 days in 2018), with a mean total stay of 7.2 days (ranging from 6.7 in 2010, 2011 and 2012 to 7.8 days in 2018) per case. Total annual in-hospital deaths ranged from 215 in 2012 to 332 deaths in 2010, while the mean in-hospital mortality rate was 15.42% (in-hospital mortality rate ranging from 13.91% in 2017 to 16.82% in 2010) (**Table 4**). Of note, these were inpatients that died with a diagnosis of HZ, but that was not necessarily the cause of death. The mortality database also provided information on deaths caused by HZ, recorded by ICD-10 from death registries in the general population, in patients ≥15 years of age from 2010 to 2018. There is no way to know if this is the basic or primary cause of death, or intervenient or concurrent one. A total of 766 deaths were recorded in this period (60% were female) with 79% occurring in those ≥65 years of age and total annual deaths ranging from 60 (in 2010) to 123 (in 2018). Over time, the total deaths were similar in males and females. The mean HZ mortality rate in patients ≥15 years of age was 0.055 deaths for every 100,000 inhabitants (ranging from 0.041 in 2010 to 0.075 deaths per year for every 100,000 inhabitants in 2018).

*Mexico*. The website of the Mexico Health Secretariat (SS) and General Health Information Office (DGIS) was used to retrieve data on emergency visits, hospital discharges, and deaths, via their 'dynamic cubes' [71]. This database is the largest in Mexico. Data on average hospital discharges and lenght of stay, as per the International Classification of Disease, Tenth Revision (ICD-10), were retrieved. It was noted that the ICD-10 category of B029 (uncomplicated HZ) had the highest number of discharges, with an average annual discharge of 292.12 and an average annual lenght of stay of 1,327.37 days from 2010 to 2017. This was followed by B022 (Herpes zoster with further nervous system involvement) with an average annual discharge of 60.5 and an average annual lenght stay of 259.6 days (**S1 Fig in S1 File**). The same trend was observed for an average lenght of stay in hospital with B029 (uncomplicated HZ) and B022 (Herpes zoster with further nervous involvement) amounting to 1,327.37 and 259.62 days, respectively (**S2 Fig in S1 File**). From 2010 to 2017, the CFR remained stable by 0.2%, except in 2013 and 2017, with a two-fold increase of 0.44% and 0.45%, respectively (**S3 Fig in S1 File**). The number of total deaths per year from 2010 to 2017, deaths ranged from 19 in 2017 to 48 in 2012 with an average 30.75 annual deaths (**S4 Fig in S1 File**). In Mexico, the annual HZ mortality rate from years 2010–2018 ranged from 0.020–0.058 per 100,000 inhabitants [71]. Population-wide incidence and mortality rates were not possible to estimate due to lack of proper registration.

*Chile*. The open-access data on hospital discharges and deaths was accessed from the official Statistics and Health Information Department' (DEIS) in Chile. Hospital discharge data was collected from 2010–2018 [72]. **Table 5** provides hospital discharges per year according to

**Table 4. Brazil: Hospitalizations caused by varicella and HZ in patients ≥65 years of age in 2010–2019.**

| Hospitalizations caused by varicella and HZ ≥65 years old | 2010 | 2011 | 2012 | 2013 | 2014 | 2015 | 2016 | 2017 | 2018 | 2019 | Total | Average |
|---|---|---|---|---|---|---|---|---|---|---|---|---|
| Total hospitalizations | 1,974 | 1,756 | 1,447 | 1,713 | 1,381 | 1,597 | 1,574 | 1,697 | 1,738 | 1,740 | 16,617 | 1,662 |
| Total in-hospital (IH) deaths | 332 | 245 | 215 | 270 | 223 | 245 | 231 | 236 | 275 | 291 | 2563 | 256 |
| IH mortality rate | 0.168 | 0.14 | 0.1486 | 0.158 | 0.162 | 0.153 | 0.147 | 0.139 | 0.158 | 0.167 | 0.1542 | 0.154 |
| Total days in hospital | 13,160 | 11,798 | 9,746 | 13,257 | 9,773 | 11,323 | 11,487 | 12,803 | 13,497 | 12,718 | 119,562 | 11,956 |
| Mean stay in hospital (days) | 6.7 | 6.7 | 6.7 | 7.7 | 7.1 | 7.1 | 7.3 | 7.5 | 7.8 | 7.3 | 7.2 | 7.2 |

HZ: Herpes Zoster.

**Table 5. Hospital discharges per year per ICD-10 category in Chile 2010–2018.**

| Condition | Code ICD- 10 | 2010 | 2011 | 2012 | 2013 | 2014 | 2015 | 2016 | 2017 | 2018 | Average |
|---|---|---|---|---|---|---|---|---|---|---|---|
| Encephalitis caused by HZ | B020 | 8 | 4 | 3 | 12 | 6 | 7 | 9 | 12 | 11 | 8.00 |
| Meningitis caused by HZ | B021 | 10 | 8 | 6 | 7 | 8 | 9 | 8 | 17 | 21 | 10.44 |
| HZ with CNS involvement | B022 | 45 | 61 | 84 | 40 | 46 | 40 | 35 | 56 | 51 | 50.89 |
| Ocular HZ | B023 | 49 | 38 | 55 | 38 | 57 | 40 | 29 | 59 | 82 | 49.67 |
| Disseminated HZ | B027 | 8 | 14 | 15 | 17 | 21 | 27 | 27 | 31 | 34 | 21.56 |
| HZ with other complications | B028 | 43 | 38 | 43 | 49 | 54 | 55 | 62 | 67 | 66 | 53.00 |
| Uncomplicated HZ | B029 | 273 | 281 | 252 | 264 | 253 | 235 | 202 | 225 | 229 | 246.00 |
| PHN | G530 | 0 | 0 | 0 | 0 | 3 | 2 | 0 | 0 | 1 | 2.00 |
| **Total** | | **436** | **444** | **458** | **427** | **448** | **415** | **372** | **467** | **495** | **440.22** |

CNS: Central nervous system; HZ: Herpes zoster; PHN: Post-herpetic neuralgia, ICD-10: International Classification of Disease, Tenth Revision.

ICD-10 category from 2010 to 2018. Uncomplicated HZ (B029) had an average hospital discharges of 246 per year during 2010 and 2018. The rest of the ICD-10 categories ranged from 8.00 to 53.00 discharges on average per year. As for Mexico, reliable population-wide incidence and mortality rates are difficult to estimate for Chile due to under-reporting of cases.

Total hospital discharges and discharges associated with the conditions for the relevant period remained relatively stable over time. The average number of hospital discharges per ICD-10 category are shown in **Table 5** and **S5 Fig in S1 File**. Average annual in-hospital stays per ICD-10 category from the period of 2010 to 2018 show that encephalitis caused by herpes zoster (B020) required the highest number of in-hospital days for the period of analysis (14.5 days on average) with the rest varying from 3.61 to 8.91 days (**S6 Fig and S4 Table in S1 File**). The average lenght of in-hospital stay as per the ICD-10 category from 2010–2018 for uncomplicated herpes zoster is 6.37 days.

The CFR ranged from 0% for OHZ and uncomplicated herpes zoster to 2.39% for DHZ in 2010–2018 (**S5 Table in S1 File**). Although the absolute number of deaths is low, it is important to note that most deaths occur in older age groups for years 2010–2017 as reported by DEIS, Chile (**S6 Table in S1 File**) [72].

*Argentina*. A data search was conducted on the burden of disease for HZ in the Health Statistics and Information Office (DEIS, Dirección de Estadística e Información en Salud) from the National Department of Health [73]. The HZ mortality rate was calculated as the number of deaths per year per 100,000 inhabitants and analyzed for patients over 15 years old for the period of 2010 to 2018. The HZ mortality rate ranged from 0.0093–0.017, being 0–0.018 in males and 0.0047–0.027 in females. In patients over 80 years old, mortality was higher, ranging from 0.105–0.448 (**Fig 3**).

A large variation in HZ mortality rates was documented for LAC [28]. However, in all countries considered, mortality rates consistently increased with age. For instance, in Argentina mortality rate in males of 65–69 years was 0.034 (0.012–0.06) and increased to 2.12 (1.1–5.25) for males above 95 years of age (2017). In Bolivia, the mortality rate increased from 0.48 (0.0034–1.51) in 65–69 years old males, to 42.9 (2.54–138.14) in males over 95 years. It was observed that mortality was higher in females across all age groups and countries. In 2017, in the age group 85–89 years of age, the mortality rate in males in Brazil, Mexico, and Panama, was 1.12 (0.39–1.76), 1.45 (0.84–3.47), and 0.39 (0.18–0.89), respectively. In the same countries, year and age group, mortality in females was 1.43 (0.21–5.16), 1.86 (0.69–5.45), and 0.72 (0.077–5.05) respectively. The incidence rate of HZ varied from 0.0021–0.014 and was stable among different countries in the LAC region between 2010 and 2017.

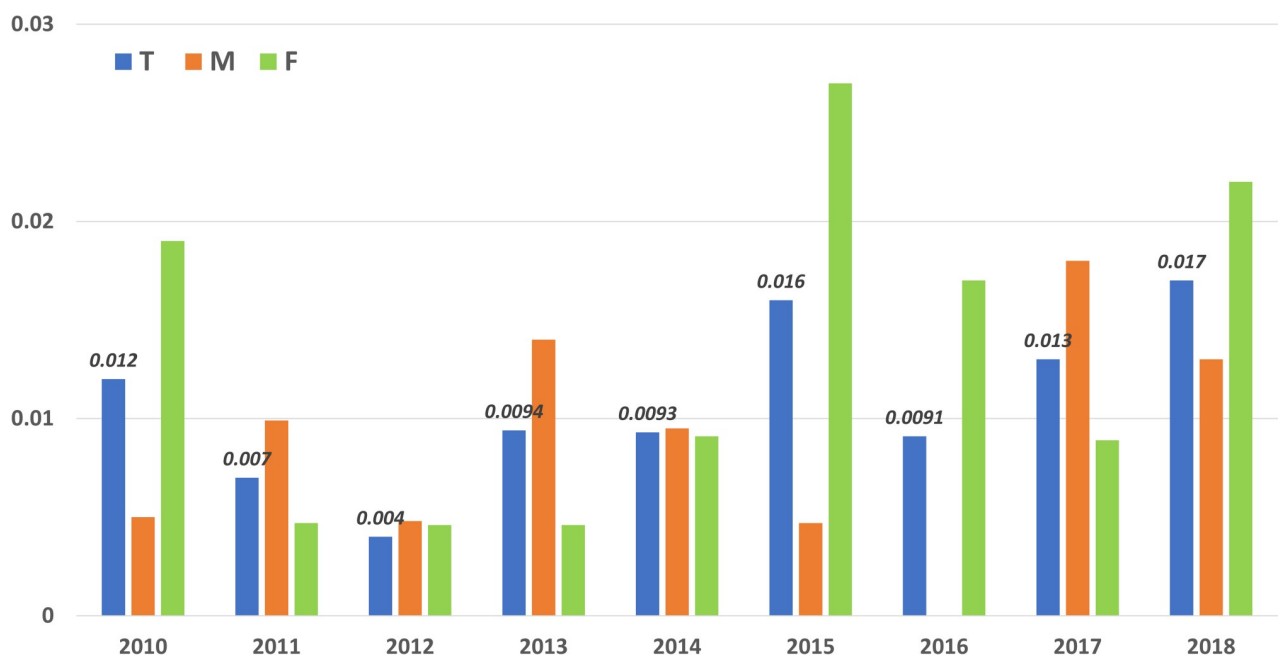

**Fig 3. Argentina: Total mortality rate per sex in patients over 15 years old in 2010–2018.** GBD Project (metrics provided by the Institute of Health Metrics).

## Discussion

This review summarizes the epidemiology and burden of HZ over the last two decades with data from 26 articles from 7 countries in the LAC region from published literature and country's official data sources. In high-risk populations, studies reported an accrued rate of incidence of 318–3,423 cases of HZ every 100,000 inhabitants per year of follow-up, with an incidence density ranging from 6.4–36.5 cases every 1,000 patient-years. No representative studies were identified on the incidence of HZ on the general population; therefore, the best approximation came from a GBD model, with incidence information for LAC countries for patients over 15 years old, ranging from 1.95 (15 to 39 years of age) to 6.18 (55–89 years of age) every 1,000 people. A systematic review including 130 studies from North America, Asia and Pacific region depicted an incidence rate of 3 and 5 per 1000-person-years [21]. This is consistent with our findings where incidence of HZ significantly increased after 50 years of age and is 6–8 per 1,000 person-years at 60 years of age and 8–12 per 1,000 person-years at 80 years of age. A lower incidence in immunocompromised individuals was found in our review when comparing the findings to other studies from the United States and Africa [74,75]. This lower incidence found in the LAC region is likely to be related to under-recording of cases, a well-known problem of the region's surveillance systems.

PHN was recorded as most common complication and data vary based on the definition being used, the age of the population, and the type of study. The proportion of HZ patients developing PHN was 18% in immunocompetent patients and 22% in immunocompromised patients. This is consistent with previously published studies [21,74]. In the meta-analysis of proportions of HZ complications in two subgroups, both in immunosuppressed and in immunocompetent patients, data were collected for PHN, OHZ, secondary bacterial infection and DHZ. There were more complications in the immunosuppressed subgroup, as expected for a higher risk group. There was large heterogeneity among the studies in the meta-analysis, explained by the fact that not all studies used the same criterion to define and report

complications. Furthermore, the large differences in the number of patients among studies increased heterogeneity [42,45,59].

Hospitalizations due to HZ remained stable over time in Brazil, Mexico, and Chile, and both in Mexico and in Chile the main cause of hospitalization was uncomplicated HZ. Similarly, the in-hospital lenght of stay (number of days per year) remained stable in these three countries. In contrast, a survey in Italy on HZ hospitalized subjects pointed out average lenght of stay equal to 23 days [76]. The main cause of stay in a hospital as a result of HZ was uncomplicated HZ in Mexico, and encephalitis caused by HZ in Chile. For total deaths due to HZ, Brazil had a slight and gradual increase over the period, which could be explained by population growth in age. This was not the case in Mexico and Chile, showing no upward trends. The HZ mortality rate for patients over 15 years of age also had a gradual increase over time in Brazil, which might correspond to further database recording. In Mexico and Argentina, HZ mortality rates remained relatively stable between years 2010 and 2019. Finally, both in Mexico and in Chile, the rate of in-hospital mortality remained stable, reaching peaks in 2013 and 2017 in Mexico and in 2016 in Chile [70–73].

The HZ mortality rate reported outside LAC in the general population ranges from 0.017 (Belgium, 1998–2007) to 0.465 (Sweden, 2006–2010) deaths per 100,000 person-years, with most deaths occurring after 60 years of age [21]. In Brazil from 2010 to 2018 in the population over 15 years of age, a mean HZ mortality rate of 0.055 per 100,000 people (ranging from 0.041–0.075 deaths per year every 100,000 inhabitants) was reported [70]. In Mexico, the annual HZ mortality rate from years 2010–2018 ranged from 0.020–0.058 per 100,000 inhabitants [71], while in Argentina the HZ mortality rate in those over 15 years old ranged from 0.0093–0.017 deaths per 100,000 people during the same time period [73]. For GBD, the mortality rate due to varicella and HZ in patients over 65 years of age from 2010 to 2017 varied among different countries and ranged from 0.0022–82.21 per 100,000 population, owing to the wide range of all-cause mortality seen among the countries in the region; Uruguay and Honduras documented the lowest and highest mortality rates in 2017 [28]. Also, the HZ cause-specific mortality rate in those over 15 years old (equivalent to prevalence for excessive mortality) did not vary among countries in 2010 and 2017, ranging from 0.0022–0.018 per 100,000 population [28]. The HZ mortality rates reported in this review were based on the information from ministerial reports, while the overall range was low, it increased with an increase in the age of the patient population. It is important to note that with regards to HZ mortality, data from the ministerial database had no distinction between low- and high-risk groups. Varicella vaccine in national immunization programs on LAC was included in many countries since 2010 or even earlier in the case of Costa Rica, Puerto Rico and Uruguay [77]. The introduction of mandatory vaccinations in the region has had a substantial beneficial effect on reducing varicella incidence and possibly herpes zoster. It is expected that the varicella and HZ burden will further decrease in these countries as more cohorts of children are vaccinated and herd immunity increases.

The sustained expansion of vaccination in other countries of the region is very promising, and more than half the population is currently living in countries with a universal vaccination program [78].

This is the most updated and comprehensive systematic review to date, including meta-analytic pooled proportion estimates for HZ complication, exploring not only published literature, but also grey sources and Ministry of Health relevant data. On the other hand, some limitations warrant further discussion. The incidence rates might not accurately represent the true burden of disease, due to under-reporting and lack of mandatory notification of all HZ cases in the region. In addition, higher rates of incidence among immunosuppressed patients could be related to the type of population included. Indeed, the high heterogeneity levels found

could be partly explained by different criteria to define the disease, report complications, and by the sparse number of patients in many studies. Finally, the inclusion of high-risk populations largely limit the generalizability of the findings to the general population in the LAC region.

In summary, our results demonstrate a low incidence of HZ in the general population, with a consistent increase in the rate of incidence among high-risk populations and older age groups over the last 20 years. Effective healthcare interventions such as antiviral therapy and hospitalization could prove beneficial to combat disease and treat HZ complications, especially in the high-risk population and individuals of older ages in the LAC region. Implementation of adult vaccination programs could prevent HZ disease and its complications among the most vulnerable populations in the LAC region.

## Supporting information

**S1 Checklist. PRISMA 2009 checklist.**
(DOC)

**S1 File.**
(DOCX)

## Acknowledgments

The authors would like to thank Business & Decision Life Sciences platform for editorial assistance and manuscript coordination, on behalf of GSK. Pierre-Paul Prévot coordinated manuscript development and editorial support. Amrita Oswald provided medical writing support.

## Author Contributions

**Conceptualization:** Ariel Esteban Bardach, Javier Nieto Guevara, Jorge Gómez, Agustin Ciapponi.

**Data curation:** Ariel Esteban Bardach, Carolina Palermo, Tomás Alconada, Macarena Sandoval, Darío Javier Balan, Javier Nieto Guevara, Jorge Gómez, Agustin Ciapponi.

**Formal analysis:** Ariel Esteban Bardach, Carolina Palermo, Tomás Alconada, Macarena Sandoval, Darío Javier Balan, Jorge Gómez, Agustin Ciapponi.

**Investigation:** Javier Nieto Guevara.

**Methodology:** Ariel Esteban Bardach, Jorge Gómez, Agustin Ciapponi.

**Validation:** Ariel Esteban Bardach, Carolina Palermo, Tomás Alconada, Macarena Sandoval, Darío Javier Balan, Agustin Ciapponi.

**Writing – original draft:** Ariel Esteban Bardach, Agustin Ciapponi.

**Writing – review & editing:** Ariel Esteban Bardach, Carolina Palermo, Tomás Alconada, Macarena Sandoval, Darío Javier Balan, Javier Nieto Guevara, Jorge Gómez, Agustin Ciapponi.

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
