## [Decision Letter · Decision Letter 0]

4 Jun 2021

PONE-D-21-07449

Herpes Zoster Epidemiology in Latin America: A Systematic Review and Meta-analysis

PLOS ONE

Dear Dr. Bardach,

Thank you for submitting your manuscript to PLOS ONE. After careful consideration, we feel that it has merit but does not fully meet PLOS ONE’s publication criteria as it currently stands. Therefore, we invite you to submit a revised version of the manuscript that addresses the points raised during the review process.

Both expert reviewers found your manuscript very interesting but have some questions, mainly concerning the impact of vaccination on incidence of HZ in the countries studied.

We look forward to receiving your revised manuscript.

Kind regards,

Graciela Andrei

Academic Editor

PLOS ONE

Journal Requirements:

2. Thank you for providing the following Funding Statement: 

"AB, AC, TA, CP, MS, DB reports grants from GSK during the conduct of the study.

JG and JN are employees and hold shares in GSK group of companies. GlaxoSmithKline Biologicals SA funded this study and was involved in all stages of study conduct, including analysis of the data. GlaxoSmithKline Biologicals SA also covered all costs associated with the development and publication of this manuscript."

We note that one or more of the authors is affiliated with the funding organization, indicating the funder may have had some role in the design, data collection, analysis or preparation of your manuscript for publication; in other words, the funder played an indirect role through the participation of the co-authors.

If the funding organization did not play a role in the study design, data collection and analysis, decision to publish, or preparation of the manuscript and only provided financial support in the form of authors' salaries and/or research materials, please review your statements relating to the author contributions, and ensure you have specifically and accurately indicated the role(s) that these authors had in your study in the Author Contributions section of the online submission form. Please make any necessary amendments directly within this section of the online submission form.  Please also update your Funding Statement to include the following statement: “The funder provided support in the form of salaries for authors [insert relevant initials], but did not have any additional role in the study design, data collection and analysis, decision to publish, or preparation of the manuscript. The specific roles of these authors are articulated in the ‘author contributions’ section.”

If the funding organization did have an additional role, please state and explain that role within your Funding Statement.

Please also provide an updated Competing Interests Statement declaring this commercial affiliation along with any other relevant declarations relating to employment, consultancy, patents, products in development, or marketed products, etc.  

Reviewers' comments:

Reviewer's Responses to Questions

**Comments to the Author**

1. Is the manuscript technically sound, and do the data support the conclusions?

Reviewer #1: Yes

Reviewer #2: Yes

2. Has the statistical analysis been performed appropriately and rigorously? 

Reviewer #1: Yes

Reviewer #2: I Don't Know

3. Have the authors made all data underlying the findings in their manuscript fully available?

Reviewer #1: Yes

Reviewer #2: Yes

4. Is the manuscript presented in an intelligible fashion and written in standard English?

Reviewer #1: Yes

Reviewer #2: Yes

5. Review Comments to the Author

Reviewer #1: Very well written. I would agree that bit is important to evaluate HZ in different populations, and this provides rationale for the implementation of a vaccine program in the LAC region. One thing I would like to see added is mention on if the patients in any of the studies were vaccinated?

Reviewer #2: The paper addresses a relevant and interesting topic such as the evaluation of HZ epidemiology in Latin America.

The manuscript is interesting, well written and quite complete.

There are few points that deserve a comment:

- crude numbers have been included in some parts of the manuscript; it should be better to have rates (e.g. lines 223, 225, 228)

- deaths and CFR are extensively taken into account. Authors correctly point out that these data should be considered with caution as many deaths could occur in patients with co-morbidities/high risk and could not be directly attributed to HZ. For this reason it seems that these data are described in too much detail

- the lenght of stay seems quite short in respect to what have been registered elsewhere. For example, a survey in Italy on HZ hospitalized subjects pointed out an average lenght of stay equal to 23 days. This reference could be added as well as discussed (Valente N et al: Temporal trends in herpes zoster-related hospitalizations in Italy, 2001-2013: differences between regions that have or have not implemented varicella vaccination. Aging Clin Exp Res DOI 10.10'7/s40520-017-0782-z)

It could be useful and interesting to know if Authors have evaluated or considered a changed trend in HZ incidence in relation to the eventual use of universal varicella vaccination in the considered countries. This is a relevant and amply debated issue and it could be interesting to have this point included at least in the discussion.

Besides, it could be interesting to know if any introduction of HZ vaccination is foreseen in the considered countries and with type of HZ vaccine.

6. PLOS authors have the option to publish the peer review history of their article (what does this mean?). If published, this will include your full peer review and any attached files.

Reviewer #1: No

Reviewer #2: **Yes: **Giovanni Gabutti

---

## [Author Response · Author response to Decision Letter 0]

8 Jul 2021

Reviewer #1: Very well written. I would agree that it is important to evaluate HZ in different populations, and this provides rationale for the implementation of a vaccine program in the LAC region. One thing I would like to see added is mention on if the patients in any of the studies were vaccinated? 

Thank you. None of the 23 included studies mention the vaccination status after carefully checking this. A line was inserted in the results describing this important fact. 

Reviewer #2: The paper addresses a relevant and interesting topic such as the evaluation of HZ epidemiology in Latin America. 

The manuscript is interesting, well written and quite complete. 

There are few points that deserve a comment: 

- crude numbers have been included in some parts of the manuscript; it should be better to have rates (e.g. lines 223, 225, 228) 

Thank you, we re-wrote this into frequencies. 

- deaths and CFR are extensively taken into account. Authors correctly point out that these data should be considered with caution as many deaths could occur in patients with co-morbidities/high risk and could not be directly attributed to HZ. For this reason it seems that these data are described in too much detail 

Thank you. The authors’ group feel no change is needed in this regard, as the cautionary note is included. 

- the lenght of stay seems quite short in respect to what have been registered elsewhere. For example, a survey in Italy on HZ hospitalized subjects pointed out an average lenght of stay equal to 23 days. This reference could be added as well as discussed (Valente N et al: Temporal trends in herpes zoster-related hospitalizations in Italy, 2001-2013: differences between regions that have or have not implemented varicella vaccination. Aging Clin Exp Res DOI 10.10'7/s40520-017-0782-z) 

Thank you. We added this contrast in the discussion. 

It could be useful and interesting to know if Authors have evaluated or considered a changed trend in HZ incidence in relation to the eventual use of universal varicella vaccination in the considered countries. This is a relevant and amply debated issue and it could be interesting to have this point included at least in the discussion. 

Besides, it could be interesting to know if any introduction of HZ vaccination is foreseen in the considered countries and with type of HZ vaccine. 

Thank you. We added two references regarding this important aspect. Arlant 2019 BMC PH, and Cashat et al 6th ISPOR Latam Conference. (Refs in the paper)

---

## [Decision Letter · Decision Letter 1]

27 Jul 2021

Herpes Zoster Epidemiology in Latin America: A Systematic Review and Meta-analysis

PONE-D-21-07449R1

Dear Dr. Bardach,

We’re pleased to inform you that your manuscript has been judged scientifically suitable for publication and will be formally accepted for publication once it meets all outstanding technical requirements.

Kind regards,

Graciela Andrei

Academic Editor

PLOS ONE

Additional Editor Comments (optional):

Reviewers' comments:

Reviewer's Responses to Questions

**Comments to the Author**

1. If the authors have adequately addressed your comments raised in a previous round of review and you feel that this manuscript is now acceptable for publication, you may indicate that here to bypass the “Comments to the Author” section, enter your conflict of interest statement in the “Confidential to Editor” section, and submit your "Accept" recommendation.

Reviewer #1: All comments have been addressed

Reviewer #2: All comments have been addressed

2. Is the manuscript technically sound, and do the data support the conclusions?

Reviewer #1: Yes

Reviewer #2: Yes

3. Has the statistical analysis been performed appropriately and rigorously? 

Reviewer #1: Yes

Reviewer #2: Yes

4. Have the authors made all data underlying the findings in their manuscript fully available?

Reviewer #1: Yes

Reviewer #2: Yes

5. Is the manuscript presented in an intelligible fashion and written in standard English?

Reviewer #1: Yes

Reviewer #2: Yes

6. Review Comments to the Author

Reviewer #1: (No Response)

Reviewer #2: Authors have addressed most of the previously raised points

The manuscript has been improved, is very interesting and complete

7. PLOS authors have the option to publish the peer review history of their article (what does this mean?). If published, this will include your full peer review and any attached files.

Reviewer #1: No

Reviewer #2: **Yes: **Giovanni Gabutti

---

## [Editor Report · Acceptance letter]

3 Aug 2021

PONE-D-21-07449R1 

Herpes Zoster Epidemiology in Latin America: A Systematic Review and Meta-analysis 

Dear Dr. Bardach:

I'm pleased to inform you that your manuscript has been deemed suitable for publication in PLOS ONE. Congratulations! Your manuscript is now with our production department. 

Kind regards, 

on behalf of

Dr. Graciela Andrei 

Academic Editor

PLOS ONE